# Equal Opportunity in Online Classification with Partial Feedback

**Yahav Bechavod**
Hebrew University
yahav.bechavod@cs.huji.ac.il

**Katrina Ligett**
Hebrew University
katrina@cs.huji.ac.il

**Aaron Roth**
University of Pennsylvania
aaroth@cis.upenn.edu

**Bo Waggoner**
University of Colorado
bwag@colorado.edu

**Zhiwei Steven Wu**
University of Minnesota
zsw@umn.edu

## Abstract

We study an online classification problem with partial feedback in which individuals arrive one at a time from a fixed but unknown distribution, and must be classified as positive or negative. Our algorithm only observes the true label of an individual if they are given a positive classification. This setting captures many classification problems for which fairness is a concern: for example, in criminal recidivism prediction, recidivism is only observed if the inmate is released; in lending applications, loan repayment is only observed if the loan is granted. We require that our algorithms satisfy common statistical fairness constraints (such as equalizing false positive or negative rates — introduced as "equal opportunity" in [18]) *at every round*, with respect to the underlying distribution. We give upper and lower bounds characterizing the cost of this constraint in terms of the regret rate (and show that it is mild), and give an *oracle efficient* algorithm that achieves the upper bound.[1]

## 1 Introduction

Many real-world prediction tasks in which fairness concerns arise — such as online advertising, short-term hiring, lending micro-loans, and predictive policing — are naturally modeled as online binary classification problems, but with an important twist: feedback is only received for one of the two classification outcomes. Clickthrough is only observed if the advertisement is shown; worker performance is only observed for candidates who were actually hired; those who are denied a loan never have an opportunity to demonstrate that they would have repaid; only if police troops were dispatched to a precinct they able to detect unreported crimes. Applying standard techniques for enforcing statistical fairness constraints on the *gathered data* can thus lead to pernicious feedback loops that can lead to classifiers that badly violate these constraints on the underlying distribution. This kind of failure to "explore" has been highlighted as an important source of algorithmic unfairness — for example, in predictive policing settings [26, 13, 14].

To avoid this problem, it is important to explicitly manage the exploration/exploitation tradeoff that characterizes learning in partial feedback settings, which is what we study in this paper. We ask for algorithms that enforce well-studied statistical fairness constraints across two protected populations (we focus on the "equal opportunity" constraint of [18], which enforces equalized false positive rates or false negative rates, but our techniques also apply to other statistical fairness constraints like "statistical parity" [12]). In particular, we ask for algorithms that satisfy these constraints (with

respect to the unknown underlying distribution) at every round of the learning procedure. The result is that the fairness constraints restrict how our algorithms can *explore*, not just how they can exploit, which makes the problem of fairness-constrained online learning substantially different from in the batch setting. The main question that we explore in this paper is: "*how much does the constraint of fairness impact the regret bound of learning algorithms?*"

## 1.1 Our Model and Results

In our setting, there is an unknown distribution $\mathcal{D}$ over *examples*, which are triples $(\hat{x}, a, y) \in \mathcal{X} \times \{-1, 1\} \times \{-1, 1\}$. Here $\hat{x} \in \mathcal{X}$ represents a vector of features in some arbitrary *feature space*, $a \in A = \{\pm 1\}$ is the *group* to which this example belongs (which we also call the *sensitive feature*), and $y \in \mathcal{Y} = \{\pm 1\}$ is a binary label. We write $x$ to denote a pair $(\hat{x}, a)$ – the set of all features (including the sensitive one) that the learner has access to.

In each round $t \in [T]$, our learner selects hypotheses from a *hypothesis class* $\mathcal{H}$ consisting of functions $h : \mathcal{X} \times A \to \mathcal{Y}$ recommending an action (or *label*) as a function of the features (potentially including the sensitive feature). We take the positive label to be the one that corresponds to observing feedback (hiring a worker, admitting a student, approving a loan, releasing an inmate, etc.) We allow algorithms that randomize over $\mathcal{H}$. Let $\Delta(\mathcal{H})$ be the set of probability distributions over $\mathcal{H}$. We refer to a $\pi \in \Delta(\mathcal{H})$ as a *convex combination of classifiers*.

**Definition 1.1** (False positive rate). *For a fixed distribution $\mathcal{D}$ on examples, we define the false positive rate (FPR) of a convex combination of classifiers $\pi \in \Delta(\mathcal{H})$ on group $j \in \{\pm 1\}$ to be*

$$FPR_j(\pi) = \mathbb{P}(\pi(x) = +1 | a = j, y = -1) = \mathbb{E}_{h \sim \pi}\left[ \mathbb{P}_{(x,y) \sim \mathcal{D}}(h(x) = +1 | a = j, y = -1) \right].$$

We denote the *difference* between false positive rates between populations as

$$\Delta_{FPR}(\pi) := FPR_1(\pi) - FPR_{-1}(\pi).$$

The fairness constraint we impose on our classifiers in this paper asks that false positive rates be approximately equalized across populations *at every round $t$*. Throughout, analogous results hold for false negative rates. These constraints were called *equal opportunity* constraints in [18].

**Definition 1.2** ($\gamma$-equalized rates [18]). *Fix a distribution $\mathcal{D}$. A convex combination $\pi \in \Delta(\mathcal{H})$ satisfies the $\gamma$-equalized false positive rate ($\gamma$-EFP) constraint if $|\Delta_{FPR}(\pi)| \leq \gamma$. We informally use the term $\gamma$-fair to refer to such a classifier or combination of classifiers.*

As we will see in Definition 2.1, we will actually allow our algorithm to have a tiny probability of ever breaking the fairness constraint.

**Remark 1.3.** *The sources of unfairness we deal with here are the differential abilities of models in $\mathcal{H}$ to predict on different populations (which we inherit from the batch setting), and the biased data collection inherent in online partial information settings. We use equal opportunity constraints only as a canonical example of a statistical fairness constraint and do not take the position that it is always the right one. Our techniques also apply to other constraints like statistical parity.*

Note that the fairness constraint is defined with respect to the true underlying distribution $\mathcal{D}$. One of the primary difficulties we face is that in early rounds, the learner has very little information about $\mathcal{D}$, and yet is required to satisfy the fairness constraint with respect to $\mathcal{D}$.

It is straightforward to see (and a consequence of a more general lower bound that we prove) that a $\gamma$-fair algorithm cannot in general achieve non-trivial regret to the set of $\gamma$-fair convex combinations of classifiers, because of ongoing statistical uncertainty about the fairness level for all non-trivial classifiers. Thus our goal is to minimize our regret to the $\gamma$-fair convex combination of classifiers that has the lowest classification error on $\mathcal{D}$, while guaranteeing that our algorithm only deploys convex combinations of classifiers that guarantee fairness level $\gamma'$ for some $\gamma' > \gamma$. Clearly, the optimal regret bound will be a function of the gap $(\gamma' - \gamma)$, and one of our aims is to characterize this tradeoff.

**Our results.** We show that the tradeoff achieved by the inefficient algorithm is tight by proving a lower bound in Section 4. In some sense, the computational inefficiency of the simple bandits reduction above is unavoidable, because we measure the regret of our learner with respect to 0/1

classification error, which is computationally hard to minimize, even for very simple classes $\mathcal{H}$ (see, e.g., [22, 16, 11]). However, we can still hope to give an *oracle efficient algorithm* for our problem. This approach, which is common in the contextual bandits literature, assumes access to an "oracle" which can in polynomial time solve the empirical risk minimization problem over $\mathcal{H}$ (absent fairness constraints), and is an attractive way to isolate the "hard part" of the problem that is often tractable in practice. Our main result, to which we devote the body of the paper, is to show that access to such an oracle is sufficient to give a polynomial-time algorithm for the fairness-constrained learning problem, matching the simple information theoretically optimal bounds described above. To do this, we use two tools. Our high-level strategy is to apply the oracle efficient stochastic contextual bandit algorithm from [2]. In order to do this, we need to supply it with an offline learning oracle for the set of classifiers that can with high probability be certified to satisfy our fairness constraints given the data so far. We construct an approximate oracle for this problem (given a learning oracle for $\mathcal{H}$) using the oracle-efficient reduction for offline fair classification from [1]. We need to overcome a number of technical difficulties stemming from the fact that the fair oracle that we can construct is only an approximate empirical risk minimizer, whereas the oracle assumed in [2] is exact. Moreover, the algorithm from [2] assumes a finite hypothesis class, whereas we need to obtain no regret to a continuous family of distributions over hypotheses. The final result is an oracle-efficient algorithm trading off between regret and fairness, allowing for a regret bound of $O(T^{2\alpha})$ to the best $\gamma$-fair classifier while satisfying $\gamma'$-fairness at every round, with a gap of $\gamma' - \gamma = O(T^{-\alpha})$ for $\alpha \in [0.25, 0.5]$.

## 1.2 Additional Related Work

We build on two lines of work in the fair machine learning literature. First, batch (non-online) classification under a variety of statistical fairness constraints: raw classification rates [8, 23, 15] (*statistical parity* [12]), positive predictive value [24, 9], and false positive and false negative rates [24, 9, 18] ; see [5] for more examples. Second, fair *online* classification and regression in the contextual bandit setting [20, 21, 25]. Unlike some of this prior work that demands stringent *individual* fairness constraints at every round, which requires strong realizability assumptions to avoid lower bounds [20], this paper interpolates by requiring *statistical* fairness constraints to be enforced, but they still must hold for every round. This allows much stronger positive results. Recently, [7] considered the problem of enforcing statistical fairness in online learning, but from a very different perspective. That work studies the *full information* and *adversarial* setting with false positive and error rates averaged across rounds. Here, we have partial and bandit feedback, distributional assumptions, and require statistical fairness guarantees at every round.

## 2 Additional Preliminaries

Throughout the paper, we assume $+\mathbf{1}, -\mathbf{1} \in \mathcal{H}$, where $+\mathbf{1}$ and $-\mathbf{1}$ are the two *constant* classifiers (that is, $+\mathbf{1}(x) = 1$ and $-\mathbf{1}(x) = -1$ for all $x$). In some cases, we will additionally assume $+\mathbf{a}, -\mathbf{a} \in \mathcal{H}$, where $+\mathbf{a}$ and $-\mathbf{a}$ are the identity function (and its negation) on the sensitive feature (that is, $+\mathbf{a}(\hat{x}, a) = a$ and $-\mathbf{a}(\hat{x}, a) = -a$ for all $\hat{x}, a$).

**The Online Setting:** The learner interacts with the environment as follows. For each round $t = 1, \ldots, T$, the learner chooses some convex combination $\pi_t \in \Delta(\mathcal{H})$. The environment draws $(x_t, y_t) \sim \mathcal{D}$ independently; the learner observes $x_t$. The learner labels the point $\hat{y}_t = h_t(x_t)$, where $h_t \sim \pi_t$. If $\hat{y}_t = +1$, the learner observes $y_t$; otherwise, there is no feedback for this round.

We measure a learner's performance using 0-1 loss, $\ell(\hat{y}_t, y_t) = \mathbb{1}[\hat{y}_t \neq y_t]$. Given a class of distributions $\mathcal{P}$ over $H \subseteq \mathcal{H}$ and a sequence of $T$ examples, the optimal convex combination of hypotheses from $H$ in hindsight is defined as $\pi^*(\mathcal{P}) = \operatorname{argmin}_{\pi \in \mathcal{P}} \sum_{t=1}^{T} \mathbb{E}_{h \sim \mathcal{P}}[\ell(h(x_t), y_t)]$.

A learner's (pseudo)-regret with respect to $\mathcal{P}$ is

$$\text{Regret} = \sum_{t=1}^{T} \mathbb{E}_{(x_t, y_t) \sim \mathcal{D}}[\ell(h(x_t), y_t)] - \sum_{t=1}^{T} \mathbb{E}_{\substack{(x_t, y_t) \sim \mathcal{D} \\ h \sim \pi^*(\mathcal{P})}}[\ell(h(x_t), y_t)].$$

In particular, when $\mathcal{P} = \{\pi \in \Delta(\mathcal{H}) : \pi \text{ satisfies } \gamma\text{-EFP}\}$, we call this the learner's $\gamma$-EFP regret.

Finally, we ask for online learning algorithms that satisfy the following notion of fairness:

**Definition 2.1** (A $\gamma$-EFP($\delta$) online learning algorithm). *An online learning algorithm is said to satisfy $\gamma$-EFP($\delta$) fairness (for $\delta \in [0, 1]$) if, with probability $1 - \delta$ over the draw of $\{(x_t, a_t, y_t)\}_{t=1}^{T} \sim \mathcal{D}^T$, simultaneously for all rounds $t \in [T]$: $\pi_t$ satisfies $\gamma$-EFP.*

**Cost Sensitive Classification Algorithms:** We aim to give oracle-efficient online learning algorithms — that is, algorithms that run in polynomial time per round, assuming access to an oracle which can solve the corresponding offline empirical risk minimization problem. Concretely, we assume oracles for solving *cost sensitive classification (CSC)* problems over $\mathcal{H}$, which are defined by a set of examples $x_j$ and a set of weights $c_j^{-1}, c_j^{+1} \in \mathbb{R}$ corresponding to the cost of a negative and positive classification respectively.

**Definition 2.2.** *Given an instance of a CSC problem $S = \{x_j, c_j^{-1}, c_j^{+1}\}_{j=1}^{n}$, a CSC oracle $\mathcal{O}$ for $\mathcal{H}$ returns $\mathcal{O}(S) \in \arg\min_{h \in \mathcal{H}} \sum_{j=1}^{n} c_j^{(h(x_j))}$. From these oracles, we will construct $\nu$-approximate CSC oracles that may have restricted ranges $\Pi \subseteq \Delta(\mathcal{H})$. Such oracles return $\mathcal{O}_\nu(S) = \pi \in \Pi$ such that $\mathbb{E}_{h \sim \pi}[\sum_{j=1}^{n} c_j^{(h(x_j))}] \leq \arg\min_{\pi \in \Pi} \mathbb{E}_{h \sim \pi}[\sum_{j=1}^{n} c_j^{(h(x_j))}] + \nu$.*

**From "Apple Tasting" to Contextual Bandits:** Online classification problems under the feedback model we study were first described as "Apple Tasting" problems [19]. The algorithm's loss at each round accumulates according to the following loss matrix:

$$
L = \begin{array}{c} \\ \hat{y} = +1 \\ \hat{y} = -1 \end{array} \begin{array}{c} y = +1 \quad y = -1 \\ \begin{pmatrix} 0 & 1 \\ 1 & 0 \end{pmatrix}, \end{array}
$$

but feedback is only observed for positive classifications (when $\hat{y} = +1$). This is a different feedback model than the more commonly studied *contextual bandits* setting. In that setting, the learner always get to observe the loss of the selected action (regardless of a positive or a negative classification). We will defer the formal description of contextual bandits to the appendix.

It is nevertheless straightforward to transform the apple tasting setting into the contextual bandits setting (similar observations have been previously made [4]).

**Proposition 2.3.** *Let $\mathcal{A}$ be an contextual bandits algorithm that guarantees a regret bound $R(T)$ with probability $1 - \delta$. There exists a transformation that maps feedbacks for apple tasting to feedbacks for contextual bandits such that $\mathcal{A}$ gurantees regret bound $2R(T)$ with probability $1 - \delta$ when running on the transformed feedbacks on any apple tasting instance.*

**Baseline approaches.** Given the reduction above, we can draw on standard methods from contextual bandits to solve our fair online learning problem. A simple baseline approach that is oracle-efficient is to perform "exploration-then-exploitation": the learner first "explores" by predicting $+1$ for roughly $T^{2/3}$ rounds, then "exploit" what we have learned by deploying the (empirically) best performing fair policy. This approach would guarantee a sub-optimal regret bound of $\tilde{O}(T^{\frac{2}{3}})$ to the best $\gamma$-fair classifier, while satisfying a $\gamma'$-fairness constraint at every round with a gap of $(\gamma' - \gamma) = O(T^{-\frac{1}{3}})$.

A more sophisticated approach starts with the observation (Lemma B.1) that although the set of "fair distributions over classifiers" is continuously large, the "fair empirical risk minimization" problem only has a single constraint, and so we may without loss of generality consider distributions over hypotheses $\mathcal{H}$ that have support of size 2. By an appropriate discretization, this allows us to restrict attention to a finite net of classifiers whenever $\mathcal{H}$ itself is finite. From this observation, one could employ a simple strategy to obtain an information theoretic result: Fix any parameter $\alpha \in [1/4, 1/2]$. The learner first predicts $+1$ for roughly $T^{2\alpha}$ rounds, then uses the collected data to define a set of fair policies according to the observed empirical distribution, and lastly runs the EXP4 algorithm [3, 6] over the set of fair policies. Such algorithm obtains a regret bound of $O(T^{2\alpha})$ to the best $\gamma$-fair classifier, while satisfying a $\gamma'$-fairness constraint at every round with a gap of $(\gamma' - \gamma) = O(T^{-\alpha})$. However, this algorithm needs to maintain a distribution of exponential size, and our goal is to match its regret rate with an oracle-efficient algorithm.

# 3 An Oracle-Efficient Algorithm

Our algorithm proceeds in two phases. First, during the first $T_0$ rounds, the algorithm performs *pure exploration* and always predicts $+1$ to collect labelled data. Because constant classifiers exactly equalize the false positive rates across populations, each exploration round satisfies our fairness constraint. The algorithm then use the collected data to form empirical fairness constraints, which we use to define our construction of a fair CSC oracle, given a CSC oracle unconstrained by fairness. Then, in the remaining rounds, we will run an adaptive contextual bandit algorithm that minimizes cumulative regret, while satisfying the empirical fairness constraint at every round.

We make two mild assumptions to simplify our analysis and the statement of our final bounds. First, we assume that negative examples from each of the two protected groups have constant probability mass: $\Pr[a = 1, y = -1], \Pr[a = -1, y = -1] \in \Omega(1)$. Second, we assume that the hypothesis class $\mathcal{H}$ contains the two constant classifiers and the identity function and its negation on the protected attribute: $\{+\mathbf{1}, -\mathbf{1}, +\mathbf{a}, -\mathbf{a}\} \subseteq \mathcal{H}$.

Our main theorem is as follows:

**Theorem 3.1.** *For any $\mathcal{H}$ and data distribution satisfying the two mild assumptions above, there exists an oracle-efficient algorithm that takes parameters $\delta \in [0, \frac{1}{\sqrt{T}}]$ and $\gamma \geq 0$ as input and satisfies $(\gamma + \beta)$-EFP($\delta$) fairness and has an expected regret at most $\tilde{O}(\sqrt{T} \ln(|\mathcal{H}|/\delta))$ with respect to the class of $\gamma$-EFP fair policies, where $\beta = O(\sqrt{\ln(|\mathcal{H}|/\delta)}/T^{1/4})$.*

**Remark 3.2.** *More generally, we can extend Theorem 3.1 to give an algorithm that satisfies $(\gamma + \beta)$-EFP($\delta$) for any $\beta > 0$, and achieves an expected regret at most $\tilde{O}\left( \frac{\ln(\frac{|\mathcal{H}|}{\delta})}{\beta^2} + \sqrt{T} \ln(|\mathcal{H}|/\delta) \right)$ with respect to the class of $\gamma$-EFP fair policies.*

**Remark 3.3.** *We state our theorem in what we believe is the most attractive parametric regime: when it can obtain a regret bound of $O(\sqrt{T})$. But it is straightforward, by modifying the length of the exploration round, to obtain a more general tradeoff—a regret bound of $O(T^{2\alpha})$ with respect to the set of $\gamma$-EFP fair policies, while satisfying $(\gamma + O(T^{-\alpha}))$-EFP($\delta$) fairness, for any $\alpha \in [1/4, 1/2]$. This tradeoff is tight, as we show in Section 4.*

**Algorithm.** The outline of our algorithm is as follows.
1. Label the first $T_0$ arrivals as $\hat{y}_t = 1$; observe their true labels.
2. Based on this data, construct an efficient *FairCSC oracle*. The oracle will be given a cost-sensitive classification objective. It returns an approximately-optimal convex combination $\pi$ of hypotheses subject to the linear constraint of $(\gamma + T^{-1/4})$-EFP on the empirical distribution of data. We show the algorithm can be implemented to always return a member of $\Pi$, defined to be the set of mixtures on $\mathcal{H}$ with support size two whose empirical fairness on the exploration data is at most $\gamma + \tilde{O}(T^{-1/4})$.
3. Instantiate a bandit algorithm with policy class $\Pi$. The bandit algorithm, a modification of [2], is described in detail in the next sections. In order to select its hypotheses, the bandit algorithm makes calls to the FairCSC oracle we implemented above.
4. For the remaining rounds $t > T_0$, choose labels $\hat{y}_t$ selected by the bandit algorithm and provide feedback to the bandit algorithm via the reduction given by Proposition 2.3.

**Analysis.** In the remainder of this section, we present our analysis in three main steps. First, we study the empirical fairness constraint given by the data collected during the exploration phase and give a reduction from a cost-sensitive classification problem subject to such fairness constraint to a standard cost-sensitive classification problem absent the constraint, based on [1]. We need to perform two modifications on the reduction method in [1]. First, we allow our algorithm to handle fairness constraints defined by a separate data set that is different from the one defining the cost objective. Secondly, we also provide a fair approximate CSC oracle that returns a sparse solution, a distribution over $\mathcal{H}$ with support size of at most 2. This will be useful for establishing uniform convergence.

Next, we present the algorithm run in the second phase: at each round $t > T_0$, the algorithm makes a prediction based on a randomized policy $\pi_t \in \Delta(\mathcal{H})$, which is a solution to a feasibility program given by [2]. We show how to rely on an approximate fair CSC oracle to solve this program efficiently. Consequently, we generalize the results of [2] to the setting in which the given oracle may only optimize the cost sensitive objective approximately. This may be of independent interest.

Finally, we bound the deviation between the algorithm's empirical regret and true expected regret. This in particular requires uniform convergence over the entire class of fair randomized policies, which we show by leveraging the sparsity of the fair distributions.

We now give the proof of Theorem 3.1, with forward references to needed theorems and lemmas.

*Proof of Theorem 3.1.* We set $T_0 = \Theta(\sqrt{T \ln(|\mathcal{H}|/\delta)})$. First, Lemma 3.4 shows that given our empirical EFP constraint, there exists an optimal policy of support size at most 2. Next, Lemma B.2 shows that, with probability $1 - \delta$ over arrivals $1, \ldots, T_0$, all convex combinations $\pi \in \Pi$ satisfy $\hat{\gamma}$-EFP for $\hat{\gamma} = \gamma + \beta$, $\beta = O\left(\sqrt{\ln(|\mathcal{H}|/\delta)}/T^{1/4}\right)$. It also implies that the optimal $\gamma$-fair policy is in the class. Theorem 3.5 shows that, given a CSC oracle for $\mathcal{H}$, we can implement an efficient approximate CSC oracle for this class $\Pi$. Theorem 3.11 shows that, given an approximate CSC oracle for any class, there is an efficient bandit algorithm that plays from this class and achieves expected regret $O\left(\ln\left(|\mathcal{H}|T/\delta\right)\sqrt{T}\right)$.

**Fairness:** In the first $T_0$ rounds we play $+\mathbf{1}$ which is 0-fair, and in the remaining rounds we play only policies from $\Pi$. With probability $1 - \delta$ over the exploration data, every member of $\Pi$ is $(\gamma + \beta)$-fair.

**Regret:** The algorithm's regret is at most $T_0$ plus its regret, on rounds $T_0 + 1, \ldots, T$, to the optimal policy in $\Pi$. By Proposition 2.3, this is at most twice the bandit algorithm's regret on those rounds. So our expected regret totals at most $O\left(\ln\left(|\mathcal{H}|T/\delta\right)\sqrt{T}\right)$ to the best policy in $\Pi$. With probability $1 - \delta$, $\Pi$ contains the optimal $\gamma$-fair classifier; with the remaining probability, the algorithm's regret to the best $\gamma$-fair classifier can be bounded by $T$. Choosing $\delta \leq \frac{1}{\sqrt{T}}$ gives the result. $\square$

## 3.1 Step 1: Constructing a Fair CSC Oracle From Exploration Data

Let $S_E$ denote the set of $T_0$ labeled examples $\{z_i = (x_i, a_i, y_i)\}_{i=1}^{T_0}$ collected from the initial exploration phase, and let $\mathcal{D}_E$ denote the empirical distribution over $S_E$. We will use $\mathcal{D}_E$ as a proxy for the true distribution to form an *empirical fairness constraint*. To support the learning algorithm in the second phase, we need to construct an oracle that solves CSC problems subject to the empirical fairness constraint. Formally, an instance of the *FairCSC* problem for the class $\mathcal{H}$ is given by a set of $n$ tuples $\{(x_j, c_j^{(-1)}, c_j^{(+1)})\}_{j=1}^n$ as before, along with a fairness parameter $\gamma$ and an approximation parameter $\nu$. We wish to solve the following fair CSC problem:

$$\min_{\pi \in \Delta(\mathcal{H})} \; \mathbb{E}_{h \sim \pi}\left[\sum_{j=1}^n c_j^{(h(x_j))}\right] \quad \text{such that} \quad |\Delta_{FPR}(\pi, \mathcal{D}_E)| \leq \gamma \tag{1}$$

where $\Delta_{FPR}(\pi, \mathcal{D}_E) = FPR_1(\pi, \mathcal{D}_E) - FPR_{-1}(\pi, \mathcal{D}_E)$ and each $FPR_j(\pi, \mathcal{D}_E)$ denotes the false positive rate of $\pi$ on distribution $\mathcal{D}_E$. We show a useful structural property that there always exists a small-support optimal solution; the proof appears in Appendix B.1.

**Lemma 3.4.** *There exists an optimal solution for the FairCSC that is a distribution over $\mathcal{H}$ with support size no greater than 2.*

We therefore consider the set of sparse convex combinations:

$$\Pi = \{\pi \in \Delta(\mathcal{H}) \mid \mathrm{Supp}(\pi) \leq 2, \quad |\Delta_{FPR}(\pi, \mathcal{D}_E)| \leq \gamma + \beta\}$$

and focus on algorithms that only play policies from $\Pi$ and measure their performance with respect to $\Pi$. For any $\pi \in \Pi$, we will write $\pi(h)$ to denote the probability $\pi$ places on $h$. Applying a standard concentration inequality, we can show (Lemma B.2) that each policy in $\Pi$ is also approximately fair with respect to the underlying distribution.

We provide a reduction from FairCSC problems to standard CSC problems as follows: 1) We first apply a standard transformation on the input CSC objective to derive an equivalent weighted classification problem, in which each example $j$ has importance weight $|c_j^{(-1)} - c_j^{(+1)}|$. 2) We then run the fair classification algorithm due to [1] that solves the weighted classification problem *approximately* using a polynomal number of CSC oracle calls. 3) Finally, we follow an approach similar to that of [10] to shrink the support size of the solution returned by the fair classification algorithm down to at most 2, which can be done in polynomial time.

**Theorem 3.5** (Reduction from FairCSC to CSC). *For any $0 < \nu < \gamma/2$, there exists an oracle-efficient algorithm that calls a CSC oracle for $\mathcal{H}$ at most $O(1/\nu^2)$ times and computes a solution $\hat{\pi} \in \Delta(\mathcal{H})$ that has a support size of at most 2, satisfies $\gamma$-EFP, and has total cost*

$$\underset{h \sim \hat{\pi}}{\mathbb{E}} \left[ \sum_{j=1}^{n} c_j^{h(x_j, a_j)} \right] \leq \min_{\pi \in \Pi} \underset{h \sim \pi}{\mathbb{E}} \left[ \sum_{j=1}^{n} c_j^{h(x_j, a_j)} \right] + \epsilon$$

*with $\epsilon = 4\nu \sum_{j=1}^{n} |c_j^{(-1)} - c_j^{(+1)}|$.*

### 3.2 Step 2: The Adaptive Learning Phase

**Overview of bandit algorithm.** In the second phase, rounds $t > T_0$, we utilize a bandit algorithm to make predictions. We now describe the algorithm, which closely follows the ILOVETOCONBANDITS algorithm by [2] but with important modifications that are necessary to handle approximation error in the FairCSC oracle.

At each round $t > T_0$, the bandit algorithm produces a distribution $Q_t$ over policies $\pi$. Each policy $\pi$ is a convex combination of two classifiers in $\mathcal{H}$ and satisfies approximate fairness. The algorithm then draws $\pi$ from $Q_t$, draws $h$ from $\pi$, and labels $\hat{y}_t = h(x_t)$. To choose $Q_t$, the algorithm places some constraints on $Q$ and runs a short coordinate descent algorithm to find a $Q$ satisfying those constraints. Finally, it mixes in a small amount of the uniform distribution over labels (which can be realized by mixing between $+\mathbf{1}$ and $-\mathbf{1}$). We will see that the constraints, called the feasibility program, correspond to roughly bounding the expected regret of the algorithm along with bounding the variance in regret of each possible $\pi$.

**Feasibility program.** To describe the feasibility program, we first introduce some notation. For each $t$, we will write $p_t$ to denote the probability that prediction $\hat{y}_t$ is selected by the learner, and $\ell_t$ be the incurred (contextual bandit) loss given by the transformation in Proposition 2.3.

for each policy $\pi \in \Pi$, let

$$\hat{L}_t(\pi) = \frac{1}{t} \sum_{s=1}^{t} \ell_s \frac{\Pr[\pi(x_s) = \hat{y}_s]}{p_s}, \qquad L(\pi) = \underset{(x,a,y) \sim \mathcal{D}}{\mathbb{E}} \left[ \underset{\pi}{\mathbb{E}} \left[ \mathbf{1}[\pi(x) \neq y] \right] \right]$$

denote the estimated average loss given by the *inverse propensity score* (IPS) estimator and true expected loss for $\pi$, respectively. Similarly, let

$$\widehat{\text{Reg}}_t(\pi) = \hat{L}_t(\pi) - \min_{\pi' \in \Pi} \hat{L}_t(\pi'), \qquad \text{Reg}(\pi) = L(\pi) - \min_{\pi' \in \Pi} L(\pi'),$$

denote the estimated average regret and the true expected regret. In order to bound the variance of the IPS estimators, we will ensure that the learner predicts each label with minimum probability $\mu_t$ at each round $t$. In particular, given a solution $Q$ for the program and a minimum probability parameter $\mu_t$, the learner will predict according to the mixture distribution $Q^{\mu_t}(\cdot \mid x)$ (a distribution that predicts $+\mathbf{1}$ w.p. $\mu_t$, and predicts according to $Q$ w.p. $1 - 2\mu_t$):

$$Q^{\mu_t}(\hat{y} \mid x) = \mu_t + (1 - 2\mu_t) \int_{\pi \in \Pi} Q(\pi) \Pr[\pi(x) = \hat{y}] d\pi$$

Note that this can be represented as a convex combination of classifiers from $\mathcal{H}$ since we assume that $+\mathbf{1} \in \mathcal{H}$. We define for each $\pi \in \Pi$, $b_t(\pi) = \frac{\widehat{\text{Reg}}_t(\pi)}{4(e-2)\mu_t \ln(T)}$, and also initialize $b_0(\pi) = 0$.

We describe the feasibility problem solved at each step. The approach and analysis directly follow and extend that of [2]. In that work, the first step at each round is to compute the best policy so far, which lets us compute $\widehat{\text{Reg}}_t(\pi)$ and $b_t(\pi)$ for any policy $\pi$. Here, our FairCSC oracle only computes approximate solutions, and so we can only compute regret relative to the approximately best policy so far, which leads to corresponding approximations $\widetilde{\text{Reg}}_t(\pi)$ and $\tilde{b}_t(\pi)$. Then, our algorithm solves the same feasibility program (although a few more technicalities must be handled): given history $H_t$ (in the second phase) and minimum probability $\mu_t$, find a probability distribution $Q$ over $\Pi$ such that

$$\int_{\pi \in \Pi} Q(\pi) \tilde{b}_{t-1}(\pi) d\pi \leq 4 \qquad \text{(Low regret)}$$

$$\forall \pi \in \Pi: \quad \underset{x \sim H_t}{\mathbb{E}} \left[ \frac{1}{Q^{\mu_t}(\pi(x) \mid x)} \right] \leq 4 + \tilde{b}_{t-1}(\pi) \qquad \text{(Low variance)}$$

Intuitively, the first constraint ensures that the estimated regret (based on historical data) of the solution is at most $\tilde{O}(1/\sqrt{t})$. The second constraint bounds the variance of the resulting IPS loss estimator for policies in $\Pi$, which in turn allows us to bound the deviation between the empirical regret and the true regret for each policy over time. Importantly, we impose a tighter variance constraint on policies that have lower empirical regret so far, which prioritizes their regret estimation.

To solve the feasibility program using our FairCSC oracle, we will run a coordinate descent algorithm, similar to [2] (full description in Section B.3 as Algorithm 1). The FairCSC oracle is used to identify and fix violated constraints. Via a potential argument similar to the one of [2], we can show that the algorithm halts in a small number of iterations. We will also bound the additional error in the output solution due to the approximation in the FairCSC oracle. In the following, let $\Lambda_0 = 0$ and for any $t \geq 1$,

$$\Lambda_t := \frac{\nu}{4(e-2)\mu_t^2 \ln(T)}.$$

where $\nu$ is the approximation parameter of the FairCSC oracle.

**Lemma 3.6.** *Algorithm 1 halts in a number of iterations (and oracle calls) that is polynomial in $\frac{1}{\mu_t}$, and outputs a weight vector $Q$ that is a probability distribution with the following guarantee:*

$$\int_{\pi \in \Pi} Q(\pi)(4 + b_{t-1}(\pi))d\pi \leq 4 + \Lambda_t$$

$$\forall \pi \in \Pi : \quad \mathop{\mathbb{E}}_{x \sim H_t}\left[\frac{1}{Q^{\mu_t}(\pi(x) \mid x)}\right] \leq 4 + b_{t-1}(\pi) + \Lambda_t.$$

### 3.3 Step 3: Regret Analysis

The key step in our regret analysis is to establish a tight relationship between the estimated regret and the true expected regret and show that for any $\pi \in \Pi$, $\text{Reg}(\pi) \leq 2\widehat{\text{Reg}}(\pi) + \epsilon_t$, with $\epsilon_t = \tilde{O}(1/\sqrt{t})$. The final regret guarantee then essentially follows from the guarantee of Lemma 3.6 that the estimated regret of our policy is bounded by $\tilde{O}(1/t)$ with proper setting of $\mu_t$.

To bound the deviation between $\text{Reg}(\pi)$ and $\widehat{\text{Reg}}_t(\pi)$, we need to bound the variance of our IPS estimators. Let us define the following for any probability distribution $P$ over $\Pi$, $\pi \in \Pi$,

$$V(P, \pi, \mu) := \mathop{\mathbb{E}}_{x \sim \mathcal{D}}\left[\frac{1}{P^\mu(\pi(x) \mid x)}\right] \qquad \hat{V}_t(P, \pi, \mu) := \mathop{\mathbb{E}}_{x \sim H_t}\left[\frac{1}{P^\mu(\pi(x) \mid x)}\right]$$

Recall that through the feasibility program, we can directly bound $\hat{V}_t(Q_t, \pi, \mu_t)$ for each round. However, to apply a concentration inequality on the IPS estimator, we need to bound the population variance $V(Q_t, \pi, \mu_t)$. We do that through a deviation bound between $\hat{V}_t(Q_t, \pi, \mu_t)$ and $V(Q_t, \pi, \mu_t)$ for all $\pi \in \Pi$. In particular, we rely on the sparsity on $\Pi$ and apply a covering argument. Let $\Pi_\eta \subset \Pi$ denote an $\eta$-cover such that for every $\pi$ in $\Pi$, $\min_{\pi' \in \Pi_\eta} \|\pi(h) - \pi'(h)\|_\infty \leq \eta$ for any $h \in \mathcal{H}$. Since $\Pi$ consists of distributions with support size at most 2, we can take the cardinality of $\Pi_\eta$ to be bounded by $\lceil |\mathcal{H}|^2/\eta \rceil$.

**Claim 3.7.** *Let $P$ be any distribution over the policy set $\Pi$, and let $\pi$ be any policy in $\Pi$. Then there exists $\pi' \in \Pi_\eta$ such that $|V(P, \pi, \mu) - V(P, \pi', \mu)|_\infty, |\hat{V}_t(P, \pi, \mu) - \hat{V}_t(P, \pi', \mu)|_\infty \leq \frac{\eta}{\mu(\mu+\eta)}$.*

**Lemma 3.8.** *Suppose that $\mu_t \geq \sqrt{\frac{\ln(2|\Pi_\eta|t^2/\delta)}{2t}}, t \geq 8\ln(2|\Pi_\eta|t^2/\delta)$. Then with probability $1 - \delta$,*

$$V(P, \pi, \mu_t) \leq 6.4\hat{V}_t(P, \pi, \mu_t) + 162.6 + \frac{2\eta}{\mu_t(\mu_t + \eta)}$$

Next we bound the deviation between the estimated loss and true expected loss for every $\pi \in \Pi$.

**Lemma 3.9.** *Assume that the algorithm solves the per-round feasibility program with accuracy guarantee of Lemma 3.6. With probability at least $1 - \delta$, we have for all $t \in [T]$ all policies $\pi \in \Pi$, $\lambda \in [0, \mu_t]$, and $t \geq 8\ln(2|\Pi_\eta|t^2/\delta)$,*

$$|L(\pi) - \hat{L}_t(\pi)| \leq (e-2)\lambda\left(188.2 + \frac{1}{t}\sum_{s=1}^{t}\left(6.4b_{s-1}(\pi) + 6.4\Lambda_{s-1} + \frac{2\eta}{\mu_s(\mu_s + \eta)}\right)\right) + \frac{\ln\left(\frac{|\Pi_\eta|T}{\delta}\right)}{\lambda t}$$

To bound the difference between $\text{Reg}(\pi)$ and $\widehat{\text{Reg}}_t(\pi)$, we will set $\eta = 1/T^2$, $\mu_t = \frac{3.2\ln(|\Pi_\eta|T/\delta)}{\sqrt{t}}$ the approximation parameter $\nu$ of FairCSC to be $1/T$.

**Lemma 3.10.** *Assume that the algorithm solves the per-round feasibility program with the accuracy guarantee of Lemma 3.6. With probability at least $1 - \delta$, we have for all $t \in [T]$ all policies $\pi \in \Pi$, and for all $t \geq 8\ln(2|\mathcal{H}|^2 T^3/\delta)$,*

$$\text{Reg}(\pi) \leq 2\widehat{\text{Reg}}_t(\pi) + \epsilon_t, \qquad and \qquad \widehat{\text{Reg}}_t(\pi) \leq 2\text{Reg}(\pi) + \epsilon_t$$

*with $\epsilon_t = \frac{1000\ln(|\mathcal{H}|^2 T^2/\delta)}{\sqrt{t}}$.*

**Theorem 3.11.** *The bandit algorithm, given access to an approximate-CSC oracle, runs in time polynomial in $T$ and achieves expected regret at most $O\left(\ln(|\mathcal{H}|T/\delta)\sqrt{T}\right)$.*

# 4 Lower Bound

In this section we show that the tradeoff that our algorithm exhibits between its regret bound and the "fairness gap" $\gamma' - \gamma$ (i.e. our algorithm is $\gamma'$-fair, but competes with the best $\gamma$-fair classifier when measuring regret) is optimal. We do this by constructing a lower bound instance consisting of two very similar distributions, $\mathcal{D}_1$ and $\mathcal{D}_2$ defined as a function of our algorithm's fairness target $\gamma$. Roughly, there are not enough samples to distinguish the distributions until at least $\Theta(\frac{1}{\gamma^2})$ rounds elapse, but in order to equalize false positive rates on both distributions, an algorithm must "play it safe" and incur constant regret per round during this time.

**Theorem 4.1.** *Fix any $\alpha \in (0, 0.5)$ and let $T \geq \sqrt[\alpha]{16}$. Fix any $\delta \leq 0.24$. There exists a hypothesis class $\mathcal{H}$ containing $\{\pm 1\}$ such that any algorithm satisfying a $T^{-\alpha}$-EFP($\delta$) fairness constraint has expected regret with respect to the set of 0-EFP fair policies of $\Omega\left(T^{2\alpha}\right)$.*

**Acknowledgments**

We thank Nati Srebro for a conversation leading to the question we study here. We thank Michael Kearns for helpful discussions at an early stage of this work. YB and KL were funded in part by Israel Science Foundation (ISF) grant 1044/16, the United States Air Force and DARPA under contract FA8750-16-C-0022, and the Federmann Cyber Security Center in conjunction with the Israel national cyber directorate. AR was funded in part by NSF grant CCF-1763307 and the United States Air Force and DARPA under contract FA8750-16-C-0022. ZSW was supported in part by a Google Faculty Research Award, a J.P. Morgan Faculty Award, a Mozilla research grant, and a Facebook Research Award. Part of this work was done while KL and ZSW were visiting the Simons Institute for the Theory of Computing, and BW was a postdoc at the University of Pennsylvania's Warren Center and at Microsoft Research, New York City. Any opinions, findings and conclusions or recommendations expressed in this material are those of the authors and do not necessarily reflect the views of JP Morgan, the United States Air Force and DARPA.

## Footnotes

[1]The full version of this paper is available at https://arxiv.org/abs/1902.02242.

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
