[Supplementary Material]

# A    Proof of Proposition 2.3

We briefly recall the contextual bandits setting below, for an arbitrary loss function:

---

**Online Learning in the Contextual Bandits Setting**

**for** $t = 1, ..., T$ **do**

    Learner chooses a convex combination $\pi_t \in \Delta(\mathcal{H})$.

    Environment draws $(x_t, y_t) \sim \mathcal{D}$ independently, learner observes $x_t$.

    Learner labels the point $\hat{y}_t = h_t(x_t)$, where $h_t \sim \pi_t$.

    Learner observes loss $\ell(\hat{y}_t, y_t) \in [0, 1]$.

---

*Proof of Proposition 2.3.* Consider the following transformed loss matrix:

$$\tilde{L} = \begin{matrix} \\ \hat{y} = +1 \\ \hat{y} = -1 \end{matrix} \overset{\displaystyle y = +1 \quad y = -1}{\begin{pmatrix} 0 & 2 \\ 1 & 1 \end{pmatrix}}$$

Given an online learning with partial feedback problem, we instantiate the bandit algorithm and always play the action it recommends. We then provide the algorithm with its feedback $\tilde{L}_{\hat{y}^t, y^t}$. This is possible because if $\hat{y}^t = +1$ then we observe $y^t$, and if not then the feedback is 1 regardless of the unobserved value of $y^t$. For a sequence of arrivals $S = \{(x^t, y^t)\}_{t=1}^T$, let $m(S)$ be the number of arrivals with $y^t = -1$. Let $L(\pi, S) = \sum_{t=1}^t L_{\hat{y}^t, y^t}$ and similarly for $\tilde{L}(\pi, S)$. Then we have for all $\pi, S$ that $\tilde{L}(\pi, S) = L(\pi, S) + m(S)$. In other words, on each round where $y^t = 1$, a prediction experiences the same loss under $L$ and under $\tilde{L}$; and on each round where $y^t = -1$, the loss is exactly one larger in the bandit setting. This difference does not depend on the prediction of the hypothesis, therefore every policy's total loss under the bandit loss is exactly $m(S)$ larger than under the original loss.

It follows that our algorithm's regret is exactly equal to the bandit algorithm's. Finally, a bookkeeping note: in order that losses be bounded in $[0, 1]$, we must repeat the above argument using $0.5\tilde{L}$ in place of $\tilde{L}$, which simply scales the bandit algorithm's regret by $0.5$ relative to our algorithm's. $\qquad\square$

# B    Missing Proofs for Section 3

## B.1    Proof of Lemma 3.4

In this subsection, we establish a useful structural property for the general problem minimizing linear loss function subject to fairness constraints. This in turn provides a proof for Lemma B.1. In particular, given a hypothesis class $\mathcal{H}$, and a training set of labelled samples $S$, vectors $a, b \in \mathbb{R}^{|\mathcal{H}|}$, consider the problem:

$$\min_{x \in \Delta(|\mathcal{H}|)} \quad a^\mathsf{T} x$$
$$\text{subject to} \quad b^\mathsf{T} x \leq \gamma$$
$$b^\mathsf{T} x \geq -\gamma$$

Note that both the problem of weighted classification or cost-sensitive classification can be viewed as an instantiation of the linear program defined above. The sparsity in the solution will be useful in our analysis.

**Theorem B.1.** *In the linear program above, there exists an optimal solution that is a distribution over $\mathcal{H}$ with support size no greater than 2.*

*Proof.* Consider the following embedding of $\mathcal{H}$ in $\mathbb{R}^2$: $\forall h \in \mathcal{H} : \phi(h) = (a_h, b_h)$. Let $A = \{\phi(h) \mid \pi \in \mathcal{H}\}$. Then the optimization problem can be written as the following problem over the convex

hull $\mathrm{conv}(A)$:

$$
\begin{aligned}
\underset{(z_1,z_2)\in\mathrm{conv}(A)}{\text{minimize}} \quad & z_1 \\
\text{subject to} \quad & z_2 \le \gamma \\
& z_2 \ge -\gamma
\end{aligned}
$$

Note there exists an optimal solution $z^*$ that lies on an edge of the polytope defined by $\mathrm{conv}(A)$. This means $z^*$ is either a vertex of $\mathrm{conv}(A)$ or can be written as a convex combination of two vertices of $\mathrm{conv}(A)$, say $z'$ and $z''$. In the former case, $z^*$ can be induced by a single hypothesis $h^* \in \mathcal{H}$, and in the latter case we know there exist $h', h'' \in \mathcal{H}$ such that $z' = \phi(h')$ and $z'' = \phi(h'')$. This means the optimal solution $z^*$ can be induced by a convex combination of hypotheses. $\qquad\square$

Then the result of Lemma B.1 follows immediately.

## B.2 Proof of Theorem 3.5

As mentioned, a standard concentration inequality immediately implies:

**Lemma B.2.** *With probability $1 - \delta$, as long as $T_0 \ge c\sqrt{T \ln(|\mathcal{H}|/\delta)}$ for some universal constant $c > 0$, we have the following. First, every policy in $\Pi$ satisfies $\gamma + 2\beta$-EFP, and second, every support-2 $\gamma$-EFP policy is in $\Pi$, for $\beta = O\left(\sqrt{\ln(|\mathcal{H}|/\delta)}/T^{1/4}\right)$.*

Recall that we collect a set of $T_0$ labeled examples $\{z_i = (x_i, a_i, y_i)\}_{i=1}^{T_0}$ during the initial exploration phase, and let $\mathcal{D}_E$ denote the corresponding empirical distribution. Recall that $\mathcal{H}$ is a hypothesis class defined over both the features and the protected group memberships. We assume that $\mathcal{H}$ contains a constant classifier (which implies that there is at least one fair classifier to be found, for any distribution). To simplify notation, we consider hypotheses that labels each example with either 0 or 1.

Suppose that we are given a cost-sensitive classification instance $(X_j, C_j^1, C_j^0)$. We would like to compute a distribution over classifiers from $\mathcal{H}$ that minimizes total cost subject to the false positive rate fairness constraint. In particular, consider the following *fair cost-sensitive classification (CSC)* problem:

$$
\min_{\pi \in \Delta(\mathcal{H})} \; \underset{h \sim \pi}{\mathbb{E}} \left[ \sum_{j=1}^{n} (C_j^1 h(X_j) + C_j^0 (1 - h(X_j))) \right] \tag{2}
$$

$$
\text{such that } \forall j \in \{\pm 1\} \qquad \mathrm{FPR}_j(\pi) - \mathrm{FPR}_{-j}(\pi) \le \gamma. \tag{3}
$$

$\mathrm{FPR}_j(\pi) = \mathbb{E}_{h \sim \pi}[\mathrm{FPR}_j(h)]$. We will write $\mathrm{OPT}_C$ to denote the objective value at the optimum for the problem, that is the minimum cost achieved by a $\gamma$-EFP policy over distribution $\mathcal{D}_E$.

Equivalently, we can consider optimizing the following objective function:

$$
\min_{\pi \in \Delta(\mathcal{H})} \; \underset{h \sim \pi}{\mathbb{E}} \left[ \sum_{j=1}^{n} W_i \, \mathbf{1}\{h(X_j) \ne Y_j\} \right]
$$

where each $W_j = |C_j^0 - C_j^1|$, $Y_j = 1$ if $C_j^0 > C_j^1$ and $Y_j = 0$ otherwise. To reduce the problem further to the same formulation of [1], we consider objective with normalized weights

$$
\min_{\pi \in \Delta(\mathcal{H})} \; \underset{h \sim \pi}{\mathbb{E}} \left[ \sum_{j=1}^{n} w_j \, \mathbf{1}\{h(X_j) \ne Y_j\} \right]
$$

such that each $w_j = W_j/(\sum_j W_j)$. To simplify notation, we will write $err(h, \mathcal{P}) = \sum_{j=1}^{n} w_j \, \mathbf{1}\{h(X_j) \ne Y_j\}$, and OPT to denote optimal objective subject to $\gamma$-EFP.

For each of the fairness constraint in (3), we will introduce a dual variable $\lambda_j \ge 0$. This allows us to define the partial Lagrangian of the problem:

$$
\mathcal{L}(\pi, \lambda) = \underset{h \sim \pi}{\mathbb{E}} [err(h, \mathcal{P})] + \sum_{j \in \{\pm 1\}} \lambda_j \left( \mathrm{FPR}_j(\pi) - \mathrm{FPR}_{-j}(\pi) - \gamma \right)
$$

By strong duality, we have

$$\text{OPT} = \min_{g\in\Delta(\mathcal{H})}\max_{\lambda\in\mathbb{R}_+^2}\mathcal{L}(g,\lambda) = \max_{g\in\Delta(\mathcal{H})}\min_{\lambda\in\mathbb{R}_+^2}\mathcal{L}(g,\lambda).$$

where OPT is the optimal objective value of the ERM problem.

[1] provide an oracle-efficient algorithm for finding a *$\nu$-approximate saddle point* $(\hat{g},\hat{\lambda})$ of the Lagrangian:

$$\mathcal{L}(\hat{\pi},\hat{\lambda}) \le \mathcal{L}(g,\hat{\lambda}) + \nu \qquad \text{for all } g\in\Delta(\mathcal{H})$$

$$\mathcal{L}(\hat{\pi},\hat{\lambda}) \ge \mathcal{L}(\hat{\pi},\lambda) - \nu \qquad \text{for all } \lambda\in\Lambda$$

In their result, the algorithm restricts the dual space to be $\Lambda = \{\|\lambda\|_1 \le B \mid \lambda\in\mathbb{R}_+^2\}$ for some sufficiently large constant $B$. Their convergence rate and approximation parameter both depend on such $C$. We show that under the assumption that $\mathcal{H}$ constains the two classifiers $\mathbf{1}[a=j]$ for all $j\in\{\pm 1\}$, it is sufficient to set $C=2$, and thus restrict the dual space to be

$$\Lambda = \{\|\lambda\|_1 \le 2 \mid \lambda\in\mathbb{R}_+^2\}$$

Consequently, we can use their algorithm to find a $\nu$-approximate saddle point with only $O\left(\frac{1}{\nu^2}\right)$ number of calls to the oracle $\text{CSC}(\mathcal{H})$.

**Lemma B.3** (Follows from Theorem 1 of [1]). *There is an oracle-efficient algorithm that computes a $\nu$-approximate saddle point for the restricted Lagrangian with $\Lambda\{\|\lambda\|_1 \le 2 \mid \lambda\in\mathbb{R}_+^2\}$, using $O\left(1/\nu^2\right)$ calls to a CSC oracle over $\mathcal{H}$.*

Moreover, the approximate saddle point provides an approximate solution for our problem.

**Lemma B.4.** *Suppose that the class $\mathcal{H}$ contains the two classifiers $\mathbf{1}[a=j]$ for all $j$ and that $(\hat{\pi},\hat{\lambda})$ is a $\nu$-approximate saddle point of the Lagrangian. Then the distribution $\hat{\pi}$ satisfies*

$$err(\hat{\pi},\mathcal{P}) \le \text{OPT} + 2\nu, \qquad \text{and} \qquad \forall j\in\{\pm 1\} \qquad \text{FPR}_j(\hat{g}) - \text{FPR}_{-j}(\hat{g}) \le \gamma + 2\nu.$$

*Proof.* Let $\pi^*$ be the optimal feasible solution for the fair ERM problem. First, by the definition of approximate saddle point, we know that

$$
\begin{aligned}
err(\hat{\pi},\mathcal{P}) &= \mathcal{L}(\hat{\pi},\mathbf{0}) \\
&\le \max_{\lambda\in\Lambda}\mathcal{L}(\hat{\pi},\lambda) \\
&\le \mathcal{L}(\hat{\pi},\hat{\lambda}) + \nu \\
&\le \min_{\pi\in\Delta(\mathcal{H})}\mathcal{L}(\pi,\hat{\lambda}) + 2\nu \\
&\le \mathcal{L}(\pi^*,\hat{\lambda}) + 2\nu = \text{OPT} + 2\nu
\end{aligned}
$$

where the equality follows from the fact that $\mathcal{L}(\pi^*,\hat{\lambda}) = \text{OPT}$.

Next, we will bound the fairness constraint violations. Suppose without loss of generality that the following fairness constraint is violated: $\text{FPR}_1(\hat{\pi}) - \text{FPR}_{-1}(\hat{\pi}) = \gamma + \alpha$ for some $\alpha \ge 0$. Let $\lambda'\in\Lambda$ such that $\lambda_1' = 2$. Then

$$\mathcal{L}(\hat{\pi},\hat{\lambda}) + \nu \ge \mathcal{L}(\hat{\pi},\lambda') = err(\hat{\pi},\mathcal{P}) + 2\alpha$$

Thus, by the assumption of approximate saddle point,

$$err(\hat{\pi},\mathcal{P}) \le \mathcal{L}(\hat{\pi},\hat{\lambda}) + \nu - 2\alpha \le \mathcal{L}(\pi^*,\hat{\lambda}) + 2\nu - \alpha = \text{OPT} + 2\nu - 2\alpha.$$

Now consider a distribution $\pi'$ that is defined as the mixture of

$$\pi' = (1-\alpha)\hat{\pi} + \alpha\mathbf{1}[a=-1].$$

This means

$$\text{FPR}_1(\pi') = (1-\alpha)\text{FPR}_1(\hat{\pi}) + \alpha\text{FPR}_1(\mathbf{1}[a=-1]) = (1-\alpha)\text{FPR}_1(\hat{\pi})$$

$$\text{FPR}_{-1}(\pi') = (1-\alpha)\text{FPR}_{-1}(\hat{\pi}) + \alpha\text{FPR}_{-1}(\mathbf{1}[a=-1]) = (1-\alpha)\text{FPR}_{-1}(\hat{\pi}) + \alpha$$

It follows that
$$\text{FPR}_1(\pi') - \text{FPR}_{-1}(\pi') = (1-\alpha)(\gamma+\alpha) - \alpha \leq \gamma$$
which implies that $\pi'$ is a feasible solution for the fair ERM problem. This implies that
$$err(\hat{\pi}, \mathcal{P}) \geq err(\pi', \mathcal{P}) - \alpha \geq \text{OPT} - \alpha$$

Thus, we have $\text{OPT} + 2\nu - 2\alpha \geq \text{OPT} - \alpha$, which implies that $\alpha \leq 2\nu$. This completes the proof. $\qquad\square$

To facilitate our analysis, we would like a solution $\hat{\pi}$ that satisfies the fairness cosntraint without any violation. To achieve this, we simply tighten the constraint by an amount of $2\nu$ and compute the $\nu$-approximate saddle point for the tightened Lagrangian, replacing $\gamma$ with $\gamma' = \gamma - 2\nu$. We also need ensure such tightening of the constraint does not severely increase the resulting error.

**Lemma B.5** (Bound on additional error from tightening.). *Suppose that $\gamma > 2\nu$. Let $\text{OPT}'$ be the objective value at the optimum for the tighted optimization problem:*

$$\min_{\pi \in \Delta(\mathcal{H})} \; \mathbb{E}_{h \sim \pi}\left[\sum_{j=1}^{n} w_j \mathbf{1}\{h(X_j) \neq Y_j\}\right]$$
$$\text{such that } \forall j \in \{\pm 1\} \qquad \text{FPR}_j(\pi) - \text{FPR}_{-j}(\pi) \leq \gamma - 2\nu$$

*Then as long as that the class $\mathcal{H}$ contains the two classifiers $\mathbf{1}[a = j]$ for both $j \in \{\pm 1\}$, $\text{OPT}' - \text{OPT} \leq 2\nu$.*

*Proof.* Let $\pi^*$ be an optimal solution to the original (un-tightened) problem. Suppose without loss of generality that the following fairness constraint is violated: $\text{FPR}_1(\hat{\pi}) - \text{FPR}_{-1}(\hat{\pi}) \geq 0$. Now consider a distribution $\pi'$ that is defined as the mixture of
$$\pi' = (1 - 2\nu)\pi^* + 2\nu \mathbf{1}[a = -1].$$
Consequently, we can write
$$\text{FPR}_1(\pi') = (1-2\nu)\text{FPR}_1(\pi^*) + 2\nu\text{FPR}_1(\mathbf{1}[a = -1]) = (1-2\nu)\text{FPR}_1(\pi^*)$$
$$\text{FPR}_{-1}(\pi') = (1-2\nu)\text{FPR}_{-1}(\pi^*) + 2\nu\text{FPR}_{-1}(\mathbf{1}[a = -1]) = (1-2\nu)\text{FPR}_{-1}(\pi^*) + 2\nu$$
It follows that
$$\text{FPR}_1(\pi') - \text{FPR}_{-1}(\pi') = (1-2\nu)(\gamma+2\nu) - 2\nu \leq \gamma$$
which implies that $\pi'$ is a feasible solution for the fair ERM problem. This implies that
$$err(\hat{\pi}, \mathcal{P}) \geq err(\pi', \mathcal{P}) - 2\nu \geq \text{OPT} - 2\nu$$
This completes the proof. $\qquad\square$

Next, we translate the approximation guarantee for the normalized weighted classification problem to the orginal cost-sensitive classification problem. This leads to our guarantee stated below.

**Lemma B.6.** *For any $0 < \nu < \gamma/2$, there exists an oracle-efficient algorithm that calls CSC oracle over $\mathcal{H}$ at most $O(1/\nu^2)$ times and computes a solution $\hat{\pi}$ that satisfies $\gamma$-EFP and has total cost*

$$\mathbb{E}_{h \sim \hat{\pi}}\left[\sum_{j=1}^{n}(C_j^1 h(X_j) + C_j^0(1 - h(X_j)))\right] \leq \text{OPT}_C + \epsilon$$

*with $\epsilon = 4\nu \sum_{j=1}^{n} |C_j^1 - C_j^0|$.*

The result of Lemma B.6 shows a computationally efficient algorithm that returns an approximate CSC solution with support size at most $O(1/\nu^2)$. Finally, we will shrink the support of the solution. To derive a sparse-support solution, we consider a linear program that computes a probability distribution over the support of $\hat{\pi}$. Then we will compute a basic solution obtain the final sparse solution (e.g. by running a variant of the ellipsoid algorithm [17]).

## B.3 Missing Details and Proofs in Section 3.2

---

**Algorithm 1:** Coordinate descent algorithm for solving the feasibility program

---

1 **Input**: history $H_t$ from previous rounds; minimum probability $\mu_t$; target accuracy parameter $\nu$

2 **Initialize**: $Q = 0$; Call $\mathrm{FairCSC}(\nu)$ to compute the policy $\pi_0$ that approximately minimizes $\hat{L}_t(\pi)$ (up to error $\nu$).

    **for** $\pi \in \Pi$ **do**

$$\text{Let} \qquad \widetilde{\mathrm{Reg}}_t(\pi) = \max\{\hat{L}_t(\pi) - \hat{L}_t(\pi_0), 0\}, \qquad \tilde{b}_t(\pi) = \frac{\widetilde{\mathrm{Reg}}_t(\pi)}{4(e-2)\mu_t \ln(T)}$$

    **end**

    **for** $\pi \in \Pi$ **do**

$$V_\pi(Q) = \mathop{\mathbb{E}}_{x \sim H_t} \left[ 1/Q^{\mu_t}(\pi(x) \mid x) \right]$$

$$S_\pi(Q) = \mathop{\mathbb{E}}_{x \sim H_t} \left[ 1/Q^{\mu_t}(\pi(x) \mid x)^2 \right]$$

$$\tilde{D}_\pi(Q) = V_\pi(Q) - (4 + \tilde{b}_{t-1}(\pi))$$

    **end**

3 **if** $\int_{\pi \in \Pi} Q(\pi)(4 + \tilde{b}_\pi)d\pi > 4$ **then**

    Replace $Q$ by $cQ$ with

$$c = \frac{4}{\int_\pi Q(\pi)(4 + \tilde{b}_{t-1}(\pi))d\pi} < 1$$

    **end**

4 **if** *calling* $\mathrm{FairCSC}(\nu)$ *for $\pi$ approximating* $\max_{\pi'} \tilde{D}_{\pi'}(Q)$*, we have* $\tilde{D}_\pi(Q) > 0$ **then**

    Add the following (positive) quantity to $Q(\pi)$ while keeping all other weights unchanged:

$$\alpha_\pi(Q) = \frac{V_\pi(Q) + \tilde{D}_\pi(Q)}{2(1 - 2\mu_t)S_\pi(Q)}$$

    **end**

5 **else**

    Halt. If the sum of the weights $Q$ is smaller than 1, let $Q$ place the remaining weight on $\pi_0$.

    Output $Q$ (note the algorithm will draw from $Q^{\mu_t}$).

    **end**

---

*Proof of lemma 3.6.* The first oracle call is used to approximately solve

$$\arg\min_\pi \hat{L}_t(\pi) = \arg\min_\pi \frac{1}{t} \sum_{s=1}^t \ell_s \frac{\Pr[\pi(x_s) = a_s]}{Q_s(a_s \mid x_s)}$$

$$= \frac{1}{\mu_t} \arg\min_\pi \sum_{s=1}^t \frac{\mu_t \ell_s}{t} \frac{\Pr[\pi(x_s) = a_s]}{Q_s(a_s \mid x_s)}$$

where, because $Q_s(a|x)$ is constrained to at least $\mu_s$ (which is decreasing in $s$), the argmin now has weights summing to at most 1. Therefore the oracle, given $\nu$, returns $\tilde{\pi}$ such that

$$\min_\pi \hat{L}_t(\pi) \leq \hat{L}_t(\tilde{\pi}) \leq \min_\pi \hat{L}_t(\pi) + \frac{\nu}{\mu_t}.$$

This implies that, for all $\pi$,

$$\widehat{\mathrm{Reg}}_t(\pi) \geq \widetilde{\mathrm{Reg}}_t(\pi) \geq \widehat{\mathrm{Reg}}_t(\pi) - \frac{\nu}{\mu_t}.$$

This gives

$$b_t(\pi) \geq \tilde{b}_t(\pi) \geq b_t(\pi) - \Lambda_t.$$

If the first condition is met and the algorithm halts, then $\int Q(\pi)(4+\tilde{b}_t(\pi))d\pi \le 4$, implying that the sum of $Q$'s weights is at most 1 (since $\tilde{b}_t(\pi) \ge 0$), and implying that $\int Q(\pi)(4+b_t(\pi))d\pi \le 4+\Lambda_t$, which is the first inequality.

Next, the oracle is called once per loop to request

$$\arg\max_\pi \tilde{D}_\pi(Q) = \arg\max_\pi \sum_{s=1}^t \frac{1}{tQ_s^{\mu_s}(a_s \mid x_s)} - (4+\tilde{b}_{t-1}(\pi))$$

There are two cases, where $\widetilde{\mathrm{Reg}}_t(\pi) = 0$ and otherwise. If 0, then we again obtain an additive $\frac{\nu}{\mu_t}$ approximation. Otherwise, after dropping terms that don't depend on $\pi$, we have

$$\arg\max_\pi \sum_{s=1}^t \frac{1}{tQ_s^{\mu_t}(a_s \mid x_s)} - \frac{\ell_s \Pr[\pi(x_s) = a_s]}{4(e-2)\mu_t t \ln(T)Q_s(a_s \mid x_s)}$$

Scaling each term by $4(e-2)\ln(T)\mu_t^2$ ensures that the sum of the weights is at most 1, implying that the approximation we get is again an additive $\Lambda_t$. So if $\pi$ is chosen by the algorithm, then $\max_{\pi'}\tilde{D}_{\pi'}(Q) \ge \tilde{D}_\pi(Q) \ge \tilde{D}_{\pi'}(Q) - \Lambda_t$. Plugging in the guarantee for $b_t$, if we let $D_\pi(Q) = V_\pi(Q) - (4+b_{t-1}(\pi))$, then we get

$$\max_{\pi^*} D_{\pi^*}(Q) + \Lambda_t \ge \tilde{D}_\pi(Q) \ge \max_{\pi^*} D_{\pi^*}(Q) - \Lambda_t.$$

So if the algorithm halts after obtaining $\pi$ from the oracle with $\tilde{D}_\pi(Q) \le 0$, then $\max_{\pi^*} D_{\pi^*}(Q) \le \Lambda_t$, which implies the second guarantee.

To show convergence of the algorithm, consider the following potential function

$$\Phi(Q) = \frac{\mathbb{E}_{H_t}\left[\mathrm{RE}(\mathcal{U}_2\|Q^{\mu_t}(\cdot \mid x))\right]}{1-2\mu_t} + \frac{\int_{\pi\in\Pi} Q(\pi)\tilde{b}_{t-1}(\pi)d\pi}{4}$$

where $\mathcal{U}_2$ denotes the uniform distribution over the two predictions and $\mathrm{RE}(p\|q)$ denotes the unnormalized relative entropy between two nonnegative vectors $p$ and $q$ in $\mathbb{R}^2$ (over the two predictions):

$$\mathrm{RE}(p\|q) = \sum_{\hat{y}\in\{\pm 1\}} \left(p_{\hat{y}}\ln(p_{\hat{y}}/q_{\hat{y}}) + q_{\hat{y}} - p_{\hat{y}}\right).$$

First, we note that any renormalization step does not increase potential, i.e. letting $c = 4/\int_\pi Q(\pi)(4+\tilde{b}_t(\pi)d\pi$, if $c < 1$ (which is equivalent to the update condition) then $\Phi(cQ) \le \Phi(Q)$. This is directly proven in Lemma 6 of [2] and we do not re-prove it. The only difference is that where we used $\tilde{b}_t(\pi)$ in the definition of $c$ and $\Phi$ [2] uses $b_{t-1}(\pi)$; but the proof does not use any property of $b_{t-1}(\pi)$ except nonnegativity.

Second, we note that a renormalization step can only occur once in a row; after that, either the algorithm halts, or the other condition ($\tilde{D}_\pi(Q) > 0$) is triggered.

Third, when the other condition is triggered, the potential decreases significantly, specifically, by at least $\frac{1}{4(1-2\mu_t)}$. This is also directly proven in Lemma 7 of [2].[2] The only difference is that the proof in that paper uses $\tilde{b}_t(\pi)$ instead of $b_{t-1}(\pi)$, which yields $\tilde{D}_\pi(Q)$ rather than $D_\pi(Q)$. However, the only property of $D_\pi(Q)$ used in the proof is $D_\pi(Q) > 0$, which is satisfied by $\tilde{D}_\pi(Q)$ as well.

The potential begins with $Q = \mathbf{0}$ at $\Phi(Q) \le \frac{\ln\frac{1}{\mu_t}}{1-2\mu_t}$, and remains nonnegative by definition, so after a polynomial number of steps, the algorithm satisfies both conditions and halts. $\square$

## B.4 Missing Proofs in Section 3.3

*Proof of Claim 3.7.* Let $\pi$ be any policy in $\Pi$. Note, in particular, that $-\mathbf{1} \in \Pi$ and let $\pi'_{-\mathbf{1}} \in \Pi_\eta$ such that

$$\min_{\pi'\in\Pi_\eta} \|-\mathbf{1} - \pi'_{-\mathbf{1}}|_\infty \le \eta$$

Then, we can see that

$$|V(P,\pi,\mu) - V(P,\pi',\mu)|_\infty \leq |V(P,-\mathbf{1},\mu) - V(P,\pi'_{-\mathbf{1}},\mu)|_\infty \leq \frac{1}{\mu} - \frac{1}{\mu+\eta} = \frac{\eta}{\mu(\mu+\eta)}$$

$$|\hat{V}_t(P,\pi,\mu) - \hat{V}_t(P,\pi',\mu)|_\infty \leq |\hat{V}_t(P,-\mathbf{1},\mu) - \hat{V}_t(P,\pi'_{-\mathbf{1}},\mu)|_\infty \leq \frac{1}{\mu} - \frac{1}{\mu+\eta} = \frac{\eta}{\mu(\mu+\eta)}$$

$\square$

The following lemma follows directly from Lemma 10 of [2].

**Lemma B.7** (Full version of Lemma 3.8). *Fix any $\mu \in [0, 1/2]$ and any $\delta \in (0,1)$. Then, with probability $1 - \delta$,*

$$V(P,\pi,\mu) \leq 6.4\hat{V}_t(P,\pi,\mu) + \frac{75(1-2\mu)\ln|\Pi_\eta|}{\mu_t^2 t} + \frac{6.3\ln(2|\Pi_\eta|^2 t^2/\delta)}{\mu_t t} + \frac{2\eta}{\mu_t(\mu_t+\eta)}$$

*for all probability distributions $P$ over $\Pi$, all $\pi \in \Pi$, and for all $t$. In particular, if*

$$\mu_t \geq \sqrt{\frac{\ln(2|\Pi_\eta|t^2/\delta)}{2t}}, t \geq 8\ln(2|\Pi_\eta|t^2/\delta)$$

*then,*

$$V(P,\pi,\mu_t) \leq 6.4\hat{V}_t(P,\pi,\mu_t) + 162.6 + \frac{2\eta}{\mu_t(\mu_t+\eta)}$$

We will make use of the following concentration inequality.

**Lemma B.8** (Freedman's inequality [6]). *Let $Z_1, ..., Z_n$ be a martingale difference sequence with $Z_i \leq R$ for all $i$. Let $V_n = \sum_{i=1}^n \mathbb{E}\left[Z_i^2 \mid Z_1, \ldots, Z_{i-1}\right]$. For any $\delta \in (0,1)$ and any $\lambda \in [0, 1/R]$, with probability at least $1 - \delta$*

$$\sum_{i=1}^n Z_i \leq (e-2)\lambda V_n + \frac{\ln(1/\delta)}{\lambda}$$

*Proof of Lemma 3.9.* By applying the Freedman's inequality and union bound, we know that with probability $1 - \delta'$, for all $t \in [T]$, $\pi \in \Pi$ and $\lambda \in [0, 1/\mu_t]$,

$$|L(\pi) - \hat{L}_t(\pi)| \leq (e-2)\lambda\left(\frac{1}{t}\sum_{s=1}^t V(Q_t, \pi, \mu_t)\right) + \frac{\ln(|\Pi_\eta|T/\delta')}{t\lambda} \tag{4}$$

By the result of Lemma 3.8, we know that with probability $1 - \delta'$, for all $P \in \Pi$, for any $\mu_t$ and $t \geq 8\ln(2|\Pi_\eta|t^2/\delta')$,

$$V(P,\pi,\mu) \leq 6.4\hat{V}_t(P,\pi,\mu_t) + 162.6 + \frac{2\eta}{\mu_t(\mu_t+\eta)} \tag{5}$$

We will condition on events of (4) and (5) for the remainder of the proof, which occurs with probability at least $1 - 2\delta'$. Then we can further rewrite

$$|L(\pi) - \hat{L}_t(\pi)| \leq (e-2)\lambda\left(\frac{1}{t}\sum_{s=1}^t\left(6.4\hat{V}_t(Q_t,\pi,\mu_t) + 162.6 + \frac{2\eta}{\mu_t(\mu_t+\eta)}\right)\right) + \frac{\ln(|\Pi_\eta|T/\delta')}{\lambda t}$$

Recall that by the accuracy guarantee of Lemma 3.6, we know for all $\pi \in \Pi$,

$$\hat{V}_t(Q_t, \pi, \mu_t) \leq 4 + b_{t-1}(\pi) + \Lambda_{t-1}$$

Thus, we can further bound

$$|L(\pi) - \hat{L}_t(\pi)|$$

$$\leq (e-2)\lambda\left(\frac{1}{t}\sum_{s=1}^t\left(6.4\left(4 + b_{s-1}(\pi) + \Lambda_{s-1}\right) + 162.6 + \frac{2\eta}{\mu_t(\mu_t+\eta)}\right)\right) + \frac{\ln(|\Pi_\eta|T/\delta')}{\lambda t}$$

$$\leq (e-2)\lambda\left(188.2 + \frac{1}{t}\sum_{s=1}^t\left(6.4b_{s-1}(\pi) + 6.4\Lambda_{s-1} + \frac{2\eta}{\mu_t(\mu_t+\eta)}\right)\right) + \frac{\ln(|\Pi_\eta|T/\delta')}{\lambda t}$$

To complete the proof, we will set $\delta' = \delta/2$. $\square$

*Proof of Lemma 3.10.* To simplify notation, let

$$C_t = (e-2)\lambda \left( 188.2 + \frac{1}{t} \sum_{s=1}^t \left( 6.4\Lambda_{s-1} + \frac{\eta}{\mu_s(\mu_s + \eta)} \right) \right) + \frac{\ln(|\Pi_\eta|T/\delta)}{t\lambda}$$

Recall that

$$\Lambda_t := \frac{\nu}{4(e-2)\mu_t^2 \ln(T)}.$$

Then as long as we have $\nu \le 1/T$ and $\eta \le 1/T^2$, we have

$$C_t \le 190(e-2)\lambda + \frac{\ln(|\Pi_\eta|T/\delta)}{t\lambda}$$

We will prove our result by induction. First, the base case holds trivially given our choice of $\epsilon_t$. Next, we will show $\mathrm{Reg}(\pi) \le 2\widehat{\mathrm{Reg}}_t(\pi) + \epsilon_t$, and $\widehat{\mathrm{Reg}}(\pi) \le 2\mathrm{Reg}_t(\pi) + \epsilon_t$ follows analogously. Observe that for any policy $\pi$, we can first decompose the regret difference as

$$\mathrm{Reg}(\pi) - \widehat{\mathrm{Reg}}_t(\pi) \le (L(\pi) - \hat{L}_t(\pi)) - (L(\pi^*) - \hat{L}_t(\pi^*))$$

where $\pi^*$ denotes the optimal policy in $\Pi$. Then using the result of Lemma 3.9, we can further bound the regret difference as follows: for any $\lambda \in [0, \mu_t]$,

$$\mathrm{Reg}(\pi) - \widehat{\mathrm{Reg}}_t(\pi)$$

$$\le \frac{6.4(e-2)\lambda}{t} \left( \sum_{s=1}^t b_{s-1}(\pi) + b_{s-1}(\pi^*) \right) + 2C_t$$

$$= \frac{1.6\lambda}{\mu_t \ln(T)t} \left( \sum_{s=1}^t \widehat{\mathrm{Reg}}_{s-1}(\pi) + \widehat{\mathrm{Reg}}_{s-1}(\pi^*) \right) + 2C_t$$

$$\le \frac{3.2\lambda}{\mu_t \ln(T)t} \left( \sum_{s=1}^t \mathrm{Reg}(\pi) + \mathrm{Reg}(\pi^*) + \epsilon_{s-1} \right) + 2C_t \qquad \text{(Induction hypothesis)}$$

$$\le \frac{3.2\lambda}{\mu_t \ln(T)t} \left( t\,\mathrm{Reg}(\pi) + \sum_{s=1}^t \epsilon_{s-1} \right) + 2C_t \qquad \qquad (\mathrm{Reg}(\pi^*) = 0)$$

$$\le \frac{3.2\lambda}{\mu_t \ln(T)} \mathrm{Reg}(\pi) + \frac{3.2\lambda}{\mu_t \ln(T)t} \left( \sum_{s=1}^t \epsilon_{s-1} \right) + 2C_t$$

We will set $\lambda = \mu_t/3.2$, which allows us to simplify the bound

$$\mathrm{Reg}(\pi) - \widehat{\mathrm{Reg}}(\pi) \le \frac{\mathrm{Reg}(\pi)}{\ln(T)} + \frac{1}{\ln(T)t} \left( \sum_{s=1}^t \epsilon_{s-1} \right) + 2C_t$$

Since $(1 - 1/\ln(T)) > 1/2$ and $\mu_t = \frac{3.2 \ln(|\Pi_\eta|T/\delta)}{\sqrt{t}}$, it follows that

$$\mathrm{Reg}(\pi) \le 2\widehat{\mathrm{Reg}}(\pi) + \frac{2}{\ln(T)t} \left( \sum_{s=1}^t \epsilon_{s-1} \right) + 4C_t$$

$$\le 2\widehat{\mathrm{Reg}}(\pi) + \frac{2}{\ln(T)t} \left( \sum_{s=1}^t \epsilon_{s-1} \right) + 4 \left( \frac{190(e-2)\ln(|\Pi_\eta|T/\delta)}{\sqrt{t}} + \frac{1}{\sqrt{t}} \right)$$

$$\le 2\widehat{\mathrm{Reg}}(\pi) + \frac{2}{\ln(T)t} \left( \sum_{s=1}^t \epsilon_{s-1} \right) + \frac{560 \ln(|\Pi_\eta|T/\delta)}{\sqrt{t}}$$

Observe that $\sum_{s=1}^t \epsilon_{s-1} = 1000(\ln(|\Pi_\eta|T/\delta)) \sum_{s=1}^{t-1} \frac{1}{s} \le 1000(\ln(|\Pi_\eta|T/\delta))\sqrt{t}$. This means

$$\mathrm{Reg}(\pi) \le 2\widehat{\mathrm{Reg}}(\pi) + \frac{2000}{\ln(T)\sqrt{t}}(\ln(|\Pi_\eta|T/\delta)) + \frac{560 \ln(|\Pi_\eta|T/\delta)}{\sqrt{t}} \le 2\widehat{\mathrm{Reg}}(\pi) + \epsilon_t$$

where the last inequality holds as long as $\ln(T) \ge 5$. $\qquad \square$

*Proof of Theorem 3.11.* The cumulative regret of the first $T_1 = 8\ln(2|\mathcal{H}|^2 T^3/\delta)$ rounds is trivially bounded by $O(\sqrt{T}\ln(|\mathcal{H}|T/\delta))$. For each of the remaining rounds, we can use Lemma 3.10 to first bound the per-round regret of the sequence of $Q_t$ as

$$\int_{\pi\in\Pi} Q_t(\pi)\text{Reg}(\pi)d\pi \leq 2\int_{\pi\in\Pi} Q_{t-1}(\pi)\widehat{\text{Reg}}(\pi)d\pi + \epsilon_{t-1}$$

By the guarantee of Lemma 3.6, we can further bound the right hand side by $(4(e-2)\mu_{t-1}\ln(T))\Lambda_{t-1} \leq O\left(\ln(|\mathcal{H}|T/\delta)/\sqrt{t}\right)$. Summing over rounds, we see that the cumulative expected regret of the sequence of $Q_t$'s is bounded by $O\left(\ln(|\mathcal{H}|T/\delta)\sqrt{T}\right)$. Finally, we need to take into account the $\mu_t$ mixture of uniform prediction at each round, which incurs an additional cumulative regret of no more than $O\left(\ln(|\mathcal{H}|T/\delta)\sqrt{T}\right)$. $\square$

# C   Lower Bound Proof

In this section, we prove theorem 4.1. We make use of a couple of standard tools:

**Lemma C.1.** *(Pinsker's Inequality) Let $\mathcal{D}_1$, $\mathcal{D}_2$ be probability distributions. Let $A$ be any event. Then:*

$$|\mathcal{D}_1(A) - \mathcal{D}_2(A)| \leq \sqrt{\frac{1}{2}KL(\mathcal{D}_1||\mathcal{D}_2)}$$

The following is a simple corollary that follows from the additivity of KL-divergence over product distributions.

**Corollary C.2.** *Let $t \in \mathbb{N}$. Consider the product distributions $\mathcal{D}_1^t$, $\mathcal{D}_2^t$. For any event $A$,*

$$\left|\mathcal{D}_1^t(A) - \mathcal{D}_2^t(A)\right| \leq \sqrt{\frac{1}{2}t \cdot KL(\mathcal{D}_1||\mathcal{D}_2)}$$

Next, for any algorithm $\mathcal{A}$, round $t$, hypothesis $h$, and distribution $\mathcal{D}$, let

$$P_t(h,\mathcal{D}) = \mathbb{P}\left[\mathcal{A} \text{ plays } h \text{ on round } t\right]$$

when given inputs from $\mathcal{D}$. We say an algorithm $(\beta,t,h)$-distinguishes distributions $\mathcal{D}_1$ and $\mathcal{D}_2$ if

$$|P_t(h,\mathcal{D}_1) - P_t(h,\mathcal{D}_2)| > \beta.$$

**Lemma C.3.** *Let $\mathcal{D}_1, \mathcal{D}_2$ be two probability distributions. No algorithm can $(\beta,t,h)$-distinguish $\mathcal{D}_1$ and $\mathcal{D}_2$ for any $h$ and $t \leq \frac{2\beta^2}{KL(\mathcal{D}_1||\mathcal{D}_2)}$.*

*Proof.* Assume for contradiction that there exists an algorithm that $(\beta,t,h)$-distinguishes $\mathcal{D}_1$ and $\mathcal{D}_2$ for some $h$ and $t \leq \frac{2\beta^2}{KL(\mathcal{D}_1||\mathcal{D}_2)}$. This defines an event $A$ such that

$$|\mathcal{D}_1^t(A) - \mathcal{D}_2^t(A)| > \beta \geq \sqrt{\frac{1}{2}tKL(\mathcal{D}_1||\mathcal{D}_2)}$$

which contradicts corollary C.2. $\square$

With these tools in hand, we are ready to prove the lower bound (Theorem 4.1):

*Proof of Theorem 4.1.* Fix any $\alpha \in (0, 0.5)$ and let $T \geq \sqrt[\alpha]{16}$. Denote $\gamma = T^{-\alpha}$. Fix any $\delta \leq 0.24$. Define the following distributions over (X,A,Y):

$\mathcal{D}_1$ given by:

|  |  | $x_1$ | $x_2$ | $x_3$ | $x_4$ |
|---|---|---|---|---|---|
| $A=-1$ | $\mathbb{P}[(x,a)]$ | 1/8 | 1/8 | 1/8 | 1/8 |
|  | $\mathbb{P}[y=1|(x,a)]$ | $0.5+4\gamma$ | $0.5-4\gamma$ | 1 | 0 |
| $A=+1$ | $\mathbb{P}[(x,a)]$ | 1/8 | 1/8 | 1/8 | 1/8 |
|  | $\mathbb{P}[y=1|(x,a)]$ | $0.5-4\gamma$ | $0.5+4\gamma$ | 1 | 0 |

$\mathcal{D}_2$ given by:

|  |  | $x_1$ | $x_2$ | $x_3$ | $x_4$ |
|---|---|---|---|---|---|
| $A = -1$ | $\mathbb{P}[(x,a)]$ | 1/8 | 1/8 | 1/8 | 1/8 |
|  | $\mathbb{P}[y = 1|(x,a)]$ | $0.5 + 4\gamma$ | $0.5 - 4\gamma$ | 1 | 0 |
| $A = +1$ | $\mathbb{P}[(x,a)]$ | 1/8 | 1/8 | 1/8 | 1/8 |
|  | $\mathbb{P}[y = 1|(x,a)]$ | $0.5 + 4\gamma$ | $0.5 - 4\gamma$ | 1 | 0 |

The available hypotheses $\mathcal{H} = \{-\mathbf{1}, +\mathbf{1}, h_1, h_2\}$ are defined as:

|  |  | $x_1$ | $x_2$ | $x_3$ | $x_4$ |
|---|---|---|---|---|---|
| $-\mathbf{1}$ | $A = -1$ | $-1$ | $-1$ | $-1$ | $-1$ |
|  | $A = +1$ | $-1$ | $-1$ | $-1$ | $-1$ |
| $+\mathbf{1}$ | $A = -1$ | $+1$ | $+1$ | $+1$ | $+1$ |
|  | $A = +1$ | $+1$ | $+1$ | $+1$ | $+1$ |
| $h_1$ | $A = -1$ | $+1$ | $-1$ | $+1$ | $-1$ |
|  | $A = +1$ | $+1$ | $-1$ | $+1$ | $-1$ |
| $h_2$ | $A = -1$ | $+1$ | $-1$ | $+1$ | $-1$ |
|  | $A = +1$ | $-1$ | $+1$ | $+1$ | $-1$ |

The performance of the hypotheses in $\mathcal{H}$ on the two distributions is given by: On $\mathcal{D}_1$:

|  | $L_{\mathcal{D}}^{0-1}(h)$ | $\Delta_{FPR}(h)$ |
|---|---|---|
| $-\mathbf{1}$ | 0.5 | 0 |
| $+\mathbf{1}$ | 0.5 | 0 |
| $h_1$ | 0.25 | $4\gamma$ |
| $h_2$ | $0.25 - 2\gamma$ | 0 |

On $\mathcal{D}_2$:

|  | $L_{\mathcal{D}}^{0-1}(h)$ | $\Delta_{FPR}(h)$ |
|---|---|---|
| $-\mathbf{1}$ | 0.5 | 0 |
| $+\mathbf{1}$ | 0.5 | 0 |
| $h_1$ | $0.25 - 2\gamma$ | 0 |
| $h_2$ | 0.25 | $4\gamma$ |

Note that on both distributions, $h_1$ and $h_2$ both have substantially lower error than the two constant classifiers, but only one of them satisfies the $\gamma$-fairness constraint — and which one of them it is depends on whether the underlying distribution is $\mathcal{D}_1$ or $\mathcal{D}_2$. Note also that one of them always satisfies a 0-fairness constraint, and so sets the benchmark for 0-EFP regret. The main fact driving our lower bound is that until the algorithm can reliably distinguish $\mathcal{D}_1$ from $\mathcal{D}_2$, it must place substantial weight on the constant classifiers, incurring high regret.

We first establish that the two distributions are hard to distinguish by showing that the KL-divergence between $\mathcal{D}_1, \mathcal{D}_2$ is bounded by $O(\gamma^2)$:

$$KL(\mathcal{D}_1||\mathcal{D}_2) = \frac{2}{8}\left(\frac{1+8\gamma}{2}\ln\left(\frac{1+8\gamma}{1-8\gamma}\right) + \frac{1-8\gamma}{2}\ln\left(\frac{1-8\gamma}{1+8\gamma}\right)\right)$$

$$= \gamma \ln\left(\frac{\frac{1+8\gamma}{1-8\gamma}}{\frac{1-8\gamma}{1+8\gamma}}\right)$$

$$= \gamma \ln\left(\left(\frac{1+8\gamma}{1-8\gamma}\right)^2\right)$$

$$= 2\gamma \ln\left(\frac{1+8\gamma}{1-8\gamma}\right)$$

$$= 2\gamma \ln\left(1 + \frac{16\gamma}{1-8\gamma}\right)$$

$$\leq 2\gamma \frac{16\gamma}{1-8\gamma}$$

$$= \frac{64\gamma^2}{2(1-8\gamma)}$$

$$\leq 64\gamma^2$$

Let $\mathcal{A}$ be a $\gamma$-EFP($\delta$) algorithm. Let $K = \frac{0.01^2}{32\gamma^2}$ (and note that, for $\alpha \in (0, 0.5)$, $K = \frac{0.01^2}{32\gamma^2} = \frac{0.01^2 T^{2\alpha}}{32} < \frac{0.01^2 T}{32} \leq T$). Let $t \leq K$ (note that the number of samples observed by time $t$ is $t' \leq t$); then by lemma C.3,

$$P_t(h_1, \mathcal{D}_2) \leq P_t(h_1, \mathcal{D}_1) + 0.01$$
$$P_t(h_2, \mathcal{D}_1) \leq P_t(h_2, \mathcal{D}_2) + 0.01$$

Observe that any convex combination $\pi$ of classifiers played under $\mathcal{D}_1$ fails to satisfy the $\gamma$-EFP constraint unless it puts weight less than $1/4$ on $h_1$. Similarly, any convex combination $\pi$ of classifiers played under $\mathcal{D}_2$ fails to satisfy the $\gamma$-EFP constraint unless it puts weight less than $1/4$ on $h_2$. Since by definition, and $\gamma$-EFP($\delta$) algorithm plays only $\gamma$-EFP hypotheses on any distribution it is played on except with probability $\delta$, we have that for all $t \in [T]$

$$P_t(h_1, \mathcal{D}_1) \leq \frac{1}{4} + \delta$$

$$P_t(h_2, \mathcal{D}_2) \leq \frac{1}{4} + \delta$$

And thus

$$P_t(h_1, \mathcal{D}_2) \leq P_t(h_1, \mathcal{D}_1) + 0.01 = 0.25 + 0.01 + \delta = 0.26 + \delta$$
$$P_t(h_2, \mathcal{D}_1) \leq P_t(h_2, \mathcal{D}_2) + 0.01 = 0.25 + 0.01 + \delta = 0.26 + \delta$$

Hence on either distribution, we have,

$$\mathbb{P}[\mathcal{A} \text{ plays } +\mathbf{1} \text{ or } -\mathbf{1} \text{ on round } t] \geq 1 - (0.25 + \delta) - (0.26 + \delta) = 0.49 - 2\delta$$

The best performing 0-EFP policy on $\mathcal{D}_1$ is $h_2$, while on $\mathcal{D}_2$ it is $h_1$. Both of these induce expected per-round loss of less than $\frac{1}{4}$. Since the expected per round loss of either $+\mathbf{1}$ or $-\mathbf{1}$ is $\frac{1}{2}$ on both distributions, if $+\mathbf{1}$ or $-\mathbf{1}$ are played with constant probability, the expected per-round regret incurred is a constant bounded away from zero. As a result, the expected 0-EFP regret of $\mathcal{A}$ is at least $\Omega(K) = \Omega\left(\frac{1}{\gamma^2}\right)$.

The result is that any $T^{-\alpha}$-EFP($\delta$) algorithm must have expected 0-EFP regret of $\Omega(T^{2\alpha})$. $\quad\square$

## Footnotes

[2]In that paper the potential function is scaled by a factor of $\tau\mu_t$ compared to here, where $\tau > 0$.