[Reviews · NeurIPS 2019]

Reviewer 1



Originality: I think the task at hand is challenging and has important applications. The proposed method is a combination of two well known techniques with a necessary modification of support size to facilitate the theoretical guarantees of regret bounds. The proposed method is well motivated and has solid theoretical analysis. Quality: I think the paper is a work in progress. There is no experimental results provided! The paper ends abruptly with no conclusion. There is clearly missing comparisons to other methods or a proposed baseline, and also discussing any weaknesses. Clarity: The paper is well-written however it ends unfinished, with no experimental section and/or conclusion. Significance: Since the empirical results are missing it's hard to judge the significance of the method. However the idea is an interesting follow up to the previous works [1,2], with good theoretical support.

Reviewer 2



Clarity: The paper was very well written, and the contribution was clear. However, I think it would help if they make it more clear that which parts are standard bandit techniques and which parts are new. For example, they could explain more why adding fairness constraint makes the problem challenging and what is the new technique they are using and how much this technique is applicable to other constraints. Originality/significance: I think this paper is the first paper to come up with an algorithm that satisfies approximate EO at each round. However, I think the comparison to related work is not very clear. As they mentioned, one can also consider either of the following two notions to enforce in contextual bandit setting: For a fixed time horizon T, in the end, the algorithm satisfies EO (average False-positive over all rounds). At each round, each individual has the same probability of FP independent of the sensitive attribute. What is special about being fair at each round? Do you have a similar regret rate with (2)? They mentioned in the related work that (2) need strong assumptions, does your algorithm provide any guarantee at the individual level? Quality: All the theorems in the paper are sound (I only check some of them in the appendix).

Reviewer 3



This paper studies the problem of online classification with partial feedback under the new constraint that the policy satisfies a fairness (equality of false positives) constraint at each round. The paper leverages careful modification of a number of technical tools to prove the O(sqrt(T)) regret with gamma + O(T^(-1/4)) fairness rate. In particular, they reduce the partial feedback setting to a contextual bandits problem, construct an approximate "fair oracle" using a modification of the reductions approach to fair classification, and then modify ILOVETOCONBANDITS to use this approximate oracle. The relevant inspiration is clearly cited, and the main contribution is combining these tools to effectively handle the fairness constraint in the online learning problem. The proposed algorithm is intuitive: accept everyone in the early rounds to gather data and use this data to determine which classifiers satisfy the constrain. Then run a "fair" contextual bandits algorithm in the later rounds on a subset of the classifiers. The analysis appears correct, though I did not do a detailed pass over some of the lemmas in appendix B.4. The matching lower bound makes the story more complete-- the analysis isn't too loose. The paper does suffer from some clarity issues. The problem and results statements are sufficiently clear, but the analysis in section 3 is not very modular and could be re-organized to focus on the key lemmas and structure of the analysis rather than exposing all of the vagaries of the modifications to the tools used in the analysis. Overall, issues of partial feedback abound in fairness applications, and this paper presents a principled procedure to achieve good regret while ensuring fairness on every round. I wouldn't be suprised to see more work in the partial feedback setting, and this paper offers a reasonable baseline and intuition for how to proceed. After rebuttal: Thanks to the authors for their response. Aside from the writing concerns in section 3, I'm still a fan of this paper. I don't think the lack of experiment section is a show-stopping flaw, and I'll argue for acceptance.

[Author Response · NeurIPS 2019]

**Experiments and Baseline**    Reviewer 1 is concerned that our paper is incomplete without experiments. We respect-fully disagree, and would like to make several points:

**1.** Our paper asks and answers a basic theoretical question: is satisfying statistical fairness constraints at every round compatible *even in principle* with obtaining the optimal $\sqrt{T}$ rate-of-regret in one-sided feedback settings? Prior to our work, this question had not been asked, and there was some reason for pessimism: Joseph et al. [20] had shown that stronger individual notions of fairness were not necessarily compatible with optimal regret bounds.

**2.** One reason experiments are useful is to establish if a method outperforms previously studied algorithms for the same problem. But as we are the first to study the problem of satisfying statistical fairness constraints in partial-information online classification settings, there is no relevant prior work to compare to. We note that prior work on fair batch classification does suggest one basic explore-then-exploit style baseline: during an "exploration round" classify all examples as positive, and then run a fair batch classification algorithm to pick a hypothesis to play for all subsequent rounds. Its not hard to see that this baseline, when optimized, obtains only a sub-optimal $T^{2/3}$ regret bound, as compared to our optimal $\sqrt{T}$ regret bound. We are happy to elaborate on this simple baseline in the paper.

**3.** Another reason for experimental results is to demonstrate that the proposed algorithm is implementable, and not simply a theoretical "upper bound". But as we note in the paper, our algorithm involves a careful combination of two works: "A Reductions Approach to Fair Classification" from ICML 2018 and "Taming the Monster" from ICML 2014. We need to make modifications to the details of these algorithms to make them work in our setting, but not in any way that would affect their running time. Both algorithms are known to run efficiently in practice, with code available.

In summary, although we agree with the reviewer that experimental work could be interesting, we disagree that it would be a major contribution of this work, or that the paper is incomplete without it. The main results of the paper cannot be established experimentally, and there is no prior work to make empirical comparisons to.

**Clarity and Discussion**    Thank you for your useful suggestions about clarity: we agree, and will try and follow your advice. In particular, we will attempt to re-work Section 3 to focus less on technical details and more on giving intuition for where and why the constraints of fairness and partial feedback require modification of existing techniques, deferring more technical details to the supplement. Briefly, in addition to converting our partial feedback setting to the more standard contextual bandits setting using the reduction from Section 2 of our paper, we face three high level obstacles: 1) We need to modify the "fair reductions" oracle to be able to handle fairness constraints arising from a *different data set* than the error objective. 2) We need to modify the "mini-monster" contextual bandit algorithm to work with an approximate oracle, rather than an exact one (because we cannot implement an exact oracle using step 1). 3) We need to modify the analysis of "mini-monster" to be able to handle our infinite hypothesis class (because our fairness constraints force our hypothesis class to consist of all convex combinations of base classifiers, even if our base class is finite). Each of these steps requires some technical work, but we agree that focusing on the basic outline will be clearer.

We will also expand discussion at your suggestion about the distinction between three fairness constraints that one could ask for, at increasing levels of strength: 1) Satisfying the equal false-positive-rate condition only asymptotically (i.e. on average, in hindsight after the completion of the algorithm). 2) Satisfying the equal false-positive-rate condition at every round (what we do in this paper). 3) Satisfying the equal false-positive-rate condition at an individual level (i.e. over only the randomness of the classifier and not the population).

Prior work had only considered condition 3 (Joseph et al. [20]), and non-trivially satisfying this condition requires realizability assumptions. Our paper asks whether we can achieve optimal regret bounds without any assumptions while still satisfying 2. In particular, this means we are also the first paper to satisfy condition 1. Since we already obtain the optimal $\sqrt{T}$ regret bound satisfying the stronger condition 2, no asymptotic improvement would be possible if we relaxed to condition 1. Condition 1 would also be meaningfully weaker, since it would offer no guarantees at any particular time step: it would allow, for example, strong discrimination at every round, so long as the direction of that discrimination varied with time.

Reviewer 2 asks if experiments would help establish if our method "ends up being fairer than some other standard method." We think the confusion here comes from the fact that we used asymptotic notation to state both the fairness and regret guarantees for our algorithm. But our algorithm satisfies a hard fairness guarantee. We will clarify this, and state our theorem using only asymptotic notation for the regret guarantee. (We also re-iterate that there are no existing standard methods for the setting we consider).

Finally, sources of unfairness: This is a good discussion to add, thank you. As with all papers that take false positive rates seriously, we implicitly assume that the labels we obtain are either correct, or have symmetric ("unbiased") errors. The sources of unfairness that we deal with relate to the differential ability of models in our class to fit both populations (which we inherit from the batch setting), and the biased data collection inherent in online partial information settings. We use EO only as a canonical example of a statistical fairness constraint, and don't take the position that it is always the right one. Our techniques also apply to other constraints like statistical parity.

[Meta-Review · NeurIPS 2019]

The majority of the reviewers agreed that the contribution is insightful and valuable, albeit in need of some work to improve the writing and comparison to related work. Therefore, I am happy to recommend acceptance.